# A software platform for real-time and adaptive neuroscience experiments

Anne Draelos[1,2,12] ✉, Matthew D. Loring[3], Maxim Nikitchenko[3], Chaichontat Sriworarat[1,2], Pranjal Gupta[2,4], Daniel Y. Sprague[1,2], Eftychios Pnevmatikakis[5], Andrea Giovannucci [6], Tyler Benster[7], Karl Deisseroth [7,8], John M. Pearson [1,2,3,4,9,13] & Eva A. Naumann [3,4,10,11,13] ✉

Current neuroscience research is often limited to testing predetermined hypotheses and post hoc analysis of already collected data. Adaptive experimental designs, in which modeling drives ongoing data collection and selects experimental manipulations, offer a promising alternative. However, such adaptive paradigms require tight integration between software and hardware under real-time constraints. We introduce *improv*, a software platform for flexible integration of modeling, data collection, analysis pipelines, and live experimental control. We demonstrate both in silico and in vivo how *improv* enables efficient experimental designs for discovery and validation across various model organisms and data types. We used *improv* to orchestrate real-time behavioral analyses, rapid functional typing of neural responses via calcium imaging, optimal visual stimulus selection, and model-driven optogenetic photostimulation of visually responsive neurons in the zebrafish brain. Together, these results demonstrate the power of *improv* to integrate modeling with data collection and experimental control to achieve next-generation adaptive experiments.

Technical progress in systems neuroscience has led to an explosion in the volume of neural and behavioral data[1–4]. New challenges in processing these large, high-dimensional data have spurred the development of new computational methods to efficiently process them[5–7]. Such theoretical and computational efforts have increasingly focused on models of population dynamics[8,9] that explicitly focus on high-dimensional neural activity[10,11]. In parallel, experiments have increasingly used complex stimuli and task structures that align more closely with those experienced in the wild[12,13] to understand how neural circuitry and dynamics govern natural behavior[14–18].

Yet this complexity has led to new challenges in experimental design. Our limitation is no longer the volume of data that can be collected but the number of hypotheses that can be tested in limited experimental time. For instance, even a few visual stimulus parameters —contrast, speed, and direction of moving gratings—imply thousands of combinations of unique stimuli, with even more for natural images. Given a fixed time budget and an increasing number of experimental

[1]Department of Biostatistics & Bioinformatics, Duke University School of Medicine, Durham, NC, USA. [2]Center for Cognitive Neuroscience, Duke University, Durham, NC, USA. [3]Department of Neurobiology, Duke University School of Medicine, Durham, NC, USA. [4]Department of Psychology & Neuroscience, Duke University, Durham, NC, USA. [5]Center for Computational Mathematics, Flatiron Institute, New York, NY, USA. [6]Joint Department of Biomedical Engineering, University of North Carolina at Chapel Hill / North Carolina State University, Chapel Hill, NC, USA. [7]Department of Bioengineering, Stanford University, Stanford, CA, USA. [8]Howard Hughes Medical Institute, Stanford University, Stanford, CA, USA. [9]Department of Electrical & Computer Engineering, Duke University, Durham, NC, USA. [10]Duke School of Medicine, Department of Cell Biology, Duke University, Durham, NC, USA. [11]Department of Biomedical Engineering, Duke University, Durham, NC, USA. [12]Present address: Departments of Biomedical Engineering and Computational Medicine & Bioinformatics, University of Michigan, Ann Arbor, MI, USA. [13]These authors jointly supervised this work: John M. Pearson, Eva A. Naumann. ✉e-mail: adraelos@umich.edu; eva.naumann@duke.edu

conditions, statistical power is likely to decrease significantly without careful experimental design[19].

But even beyond time limitations, many new questions can only be addressed when experiments can be adjusted during data acquisition. For example, behaviorally relevant neurons are widely distributed[20,21] with unknown initial identities and locations. In these cases, performing causal testing via targeted stimulation methods requires first collecting data to assess the location and function of the relevant neural populations[22,23]. Moreover, many quantities of interest can only be learned from data, including information about the organization of behavioral states[24], which behavioral variables are associated with neural activity, or which neural dynamics are most relevant to behavior[25]. By contrast, typical analyses are performed long after data acquisition, precluding any meaningful interventions that would benefit from information collected during the experiment[26]. This separation between data collection and informative analysis thus directly impedes our ability to test complex functional hypotheses that might emerge *during* the experiment.

Tighter model-experiment integration offers a potential solution: Models can speed up hypothesis testing by selecting the most relevant tests to conduct. Models can also be learned or refined continually as data is acquired. Such adaptive paradigms have been used with great success in learning features of artificially generated images that maximally excite neurons in the visual cortex[27,28] or for system identification of sensory processing models by optimizing the presented stimuli[29]. Likewise, various models of decision-making can be tested and differentiated via decoding and causally perturbing a latent task variable using moment-to-moment readouts[30]. Closed-loop or adaptive designs led to identifying performance variability through real-time auditory disruptions[31] and finding stimuli that optimally excite neurons with closed-loop deep learning[32]. These experimental design strategies all utilize models that are updated as soon as new data or test results become available.

Indeed, strategies that 'close the loop' are also essential for causal experiments that directly intervene in neural systems[16,22,26,33]. For instance, in experiments that aim to mimic endogenous neural activity via stimulation, real-time feedback can inform where or when to stimulate[22,34–39], and such stimulations are critical to revealing the functional contributions of individual neurons to circuit computations and behavior[16]. In fact, for large circuits composed of thousands of neurons, establishing fine-grained causal connections between neurons may prove infeasible without models to narrow down candidate mechanisms or circuit hypotheses in real time[40].

Thus, for testing many complex hypotheses, data analysis should not be independent from data acquisition. Yet while modern computing and new algorithms have made real-time preprocessing of large-scale recordings feasible[7,41–43], significant technical barriers have prevented their routine use. For example, many successful model-based experiments have required significant offline or cluster-based computing resources[32,44]. In addition, most existing algorithms and software are not constructed to facilitate the parallel execution and streaming analyses critical for adaptive experiments. Finally, inter-process data sharing, concurrent execution, and pipeline specification pose significant technical difficulties, and because adaptive designs vary so widely, any practical solution must be easily configurable and extensible to facilitate rapid prototyping. That is, simply allowing users to choose from a set of built-in models is insufficient: the system must allow experimenters to flexibly compose existing methods with entirely novel algorithms of their own design.

To address these challenges, we present *improv*, a modular software platform for constructing and orchestrating adaptive experiments. By carefully managing the backend software engineering of data flow and task execution, *improv* can integrate customized models, analyses, and experimental logic into data pipelines in real time without requiring user oversight. Any type of input or output data stream can be defined and integrated into the setup (e.g., behavioral or neural variables), with information centralized in memory for rapid, integrative analysis. Rapid prototyping is facilitated by allowing simple text files to define arbitrary processing pipelines and streaming analyses. In addition, *improv* is designed to be highly stable, ensuring data integrity through intensive logging and high fault tolerance. It offers out-of-the-box parallelization, visualization, and user interaction via a lightweight Python application programming interface. The result is a flexible real-time preprocessing and analysis platform that achieves active model-experiment integration in only a few lines of code.

## Results

### *improv* is a flexible and user-friendly real-time software platform

We created *improv* to easily integrate custom real-time model fitting and data analysis into adaptive experiments by seamlessly interfacing with many different types of data sources (Fig. 1a). *improv*'s design is based on a simplified version of the 'actor model' of concurrent systems[45]. In this model, each independent function of the system is the responsibility of a single actor. For example, one actor could be responsible for acquiring images from a camera, with a separate actor responsible for processing those images. Each actor is implemented as a user-defined Python class that inherits from the Actor class provided by *improv* and is instantiated inside independent processes (Supplementary Fig. 1). Actors interact directly with other actors via message passing, with messages containing keys that correspond to items in a shared, in-memory data store built atop the Plasma library from Apache Arrow[46]. Rather than directly passing gigabytes worth of images from actor to actor (e.g., from acquisition to analysis steps), the image is placed into the shared data store, after which a message with the image's location is passed to any actor requiring access. Thus, communication overhead and data copying between processes is minimized (Fig. 1b). At a higher level, pipelines are defined by processing steps (actors) and message queues, which correspond to nodes and edges in a directed graph (Fig. 1c). This concurrent framework also allows *improv* to ignore any faults in individual actors and maintain processing performance for long timescales without crashing or accumulating lag (Supplementary Fig. 2).

### Real-time modeling of neural responses

Designed for flexibility, *improv* facilitates a wide class of experiments involving real-time modeling, closed-loop control, and other adaptive designs. To test these capabilities in silico, we first benchmarked its performance on a prerecorded two-photon calcium imaging data set. Using raw fluorescence images streamed from disk at the rate of original data acquisition (3.6 Hz), we simulated an experiment in which larval zebrafish were exposed to a sequence of visual whole field motion stimuli. The *improv* pipeline acquired the images, preprocessed them, analyzed the resulting deconvolved fluorescence traces, estimated response properties and functional connectivity of identified neurons, and displayed images and visuals of the results in a graphical user interface (GUI) (Fig. 2a). Images were originally acquired via two-photon calcium imaging of 6-day old larval zebrafish expressing the genetically encoded calcium indicator GCaMP6s in almost all neurons ("Methods"). Simultaneously, repetitions of visual motion stimuli, square-wave gratings moving in different directions, were displayed to the fish from below (Fig. 2b). These two data streams were sent to *improv* and synchronized via alignment to a common reference frame across time (Fig. 2c, d).

Next, calcium images were preprocessed with an actor ('Caiman Online') that used the sequential fitting function from the CaImAn library[7] to extract each neuron's spatial location (ROI) and associated neural activity traces (fluorescence and spike estimates) across time (Fig. 2e; Supplementary Fig. 3). The visual stimuli and fluorescence traces were then used to compute each neuron's response to motion direction, providing streaming and continually updated directional

tuning curves. Additionally, within a separate 'LNP Model' actor, we fit a version of a linear-nonlinear-Poisson (LNP) model[47], a widely used statistical model for neural firing (Fig. 2f; Supplementary Fig. 4). Here, in place of the entire data set, we used a sliding window across the most recent 100 frames and stochastic gradient descent to update the model parameters after each new frame of data was acquired. This model also scaled well to populations of thousands of neurons, allowing us to obtain up-to-the-moment estimates of model parameters across the brain, including circuit-wide functional connections among neurons. In testing, this online model fit converged quickly towards the value obtained by fitting the model offline using the entire data set. As a result, our replication experiment had the option of stopping early without needing to present each stimulus 5-10 times.

Finally, we constructed a GUI that displayed neural functional responses and connectivity maps in real time and offered interactive controls (Fig. 2g). While fully automating experiments could, in principle, enable more efficient experiments, it also remained important to provide status metrics and raw data streams that allowed for experimenter oversight. Here, we used the cross-platform library PyQt[48] to implement a 'Data Visualization' actor as a frontend display, visualizing raw and processed data in real time using *improv* as the backend controller. All plots were updated as new data were received, up to 60 times per second, providing users with up-to-the-minute visual feedback (Supplementary Video 1). In this way, *improv* can easily integrate incoming data with models to produce both visualizations and model-based functional characterizations in real time, with the benefit of early stopping, saving valuable experimental time.

## Concurrent neural and behavioral analyses

We next demonstrate how *improv* can be used for streaming, simultaneous analysis of neural data and behavior in real time. To do so, we reproduced an analysis from Musall et al.[49] in which features from

mouse behavioral videos acquired at 30 frames per second were used to predict neural activity acquired as two-photon calcium imaging data in the dorsal cortex. As model variables from their behavioral video data had the most power for predicting single-neuron activity, we focused our replication solely on the video data rather than any other task information. Importantly, this study identified unstructured movements, which are generally not known ahead of time, as being the strongest predictors of neural activity, suggesting that identifying those significant behavioral metrics during the experiment would generate new hypotheses to test behavioral-brain links online.

To demonstrate this possibility, *improv* ingested simultaneously recorded video of a mouse and two-photon fluorescence traces. We implemented a streaming dimension reduction method to reduce each ($240 \times 320$) video frame down to ten dimensions, used ridge regression to predict the neural activity from the low-dimensional behavioral video factors, and visualized the results (Fig. 3a). Here, we used our recently developed form of streaming dimension reduction, proSVD, to identify a stable low-dimensional representation of the video data[50]. Within one minute, proSVD found a subspace of behavioral features that were stable across time, one suitable to serve as a reduced data representation in the subsequent regression model (Fig. 3b). We next used an implementation of streaming ridge regression[51] to predict neural data from the proSVD-derived features. We found that our identified regression coefficients β also converged quickly on the order of minutes (Fig. 3c; Supplementary Fig. 5).

To gain insight into what low-dimensional behavioral features were most significant for predicting neural activity, we visualized this dimension-reduced data, plotting the first two proSVD dimensions (Fig. 3d, orange trajectory). Simultaneously, as Musall et al., we visualized the identified effects by overlaying the weighted regression coefficients onto the original behavioral video, which highlighted the relevant regions of the image used in predicting neural activity (Fig. 3d;

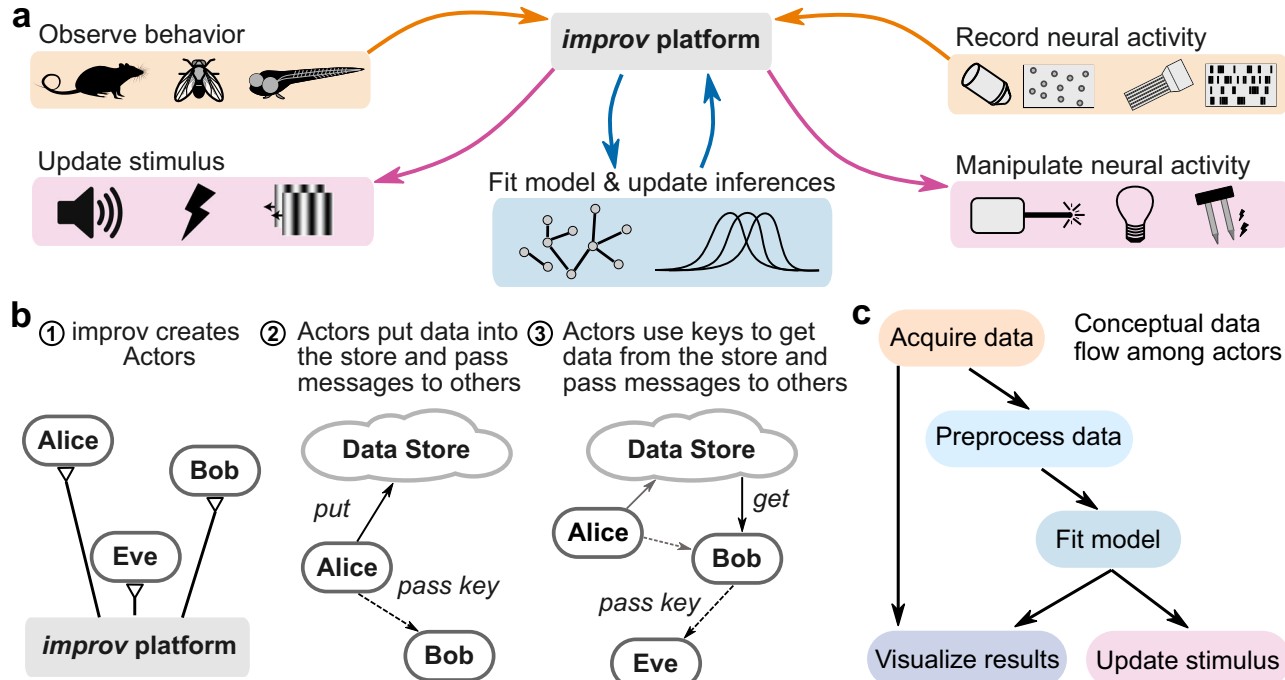

**Fig. 1 | Design architecture of *improv*. a** Schematic for possible use cases of *improv*, enabling real-time data collection from multiple sources (orange), modeling and analyzing these data via user-defined code (blue), and manipulating experimental variables (magenta). *improv* orchestrates input and output data streams independently and asynchronously. **b** Schematic for the actor model. (1) *improv* creates and manages actors as separate, concurrent processes. (2) Actors can access a shared data store and pass messages to other actors. (3) Actors send only addresses of data items, minimizing data copies. **c** Example actor graph, a pipeline that acquires neural data, preprocesses them, analyzes the resulting neural activity, suggests the next stimulus to present, and visualizes the result. Actors correspond to nodes in the processing pipeline, and arrows indicate logical dependencies between actors.

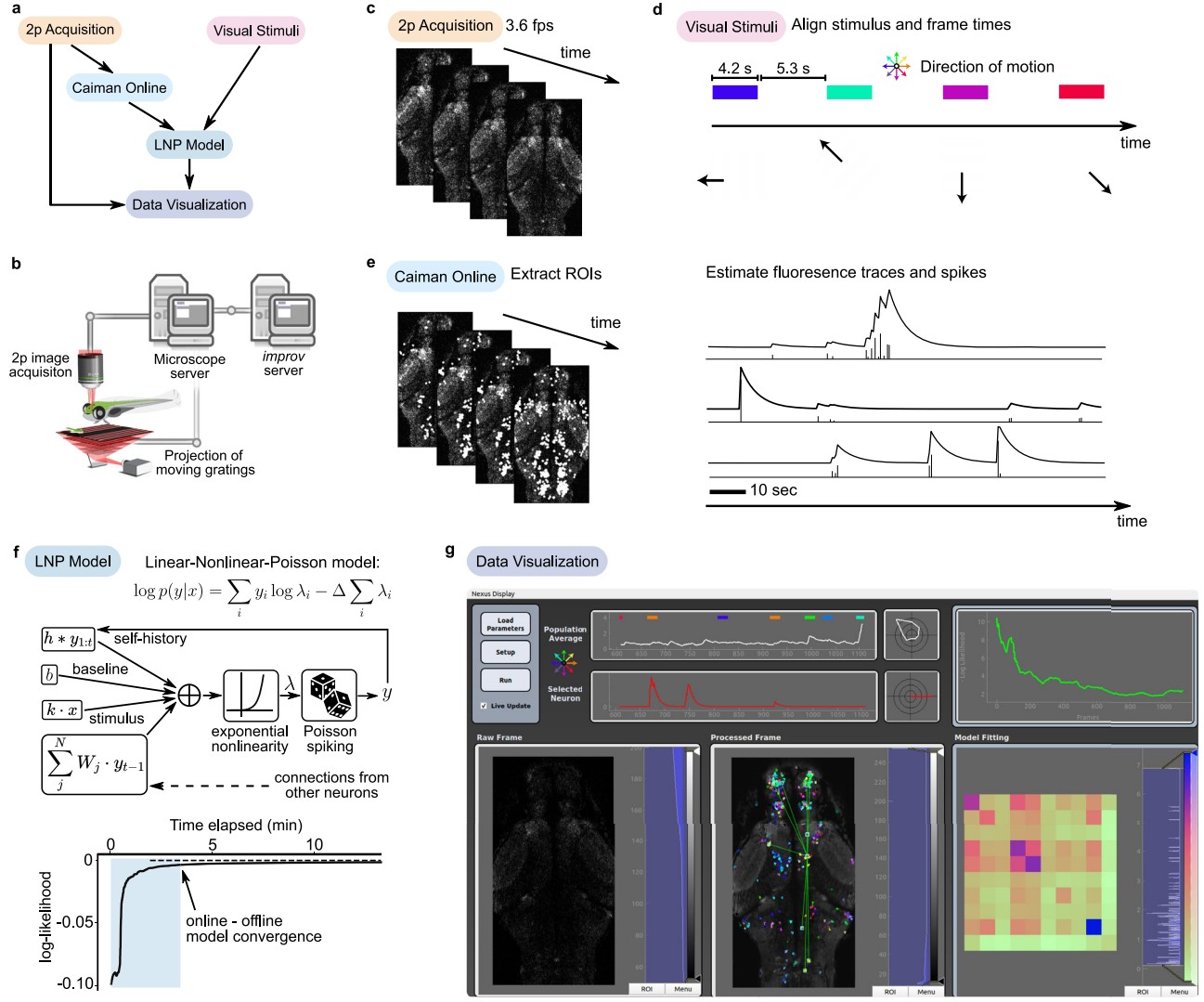

**Fig. 2 | *improv* provides streaming model-based characterization of neural function. a** Diagram showing the conceptual flow of data among all actors in the pipeline. Fluorescence images and visual stimuli data were acquired, preprocessed, fed into the model, and visualized, all in real-time. **b** Schematic of calcium imaging in zebrafish. An acquisition computer acquired fluorescence images and controlled the projection of visual stimuli, and a second networked computer running *improv* received data for processing in real time. **c** The '2p Acquisition' actor was responsible only for sending images from the two-photon microscope to the *improv* computer, one image at a time (3.6 frames/s). **d** The 'Visual Stimuli' actor broadcast information about the stimulus status and displayed visual stimuli. Stimuli were interleaved moving (4.2 s) and stationary (5.3 s) square wave gratings drifting in eight directions (arrow wheel). **e** Each image was streamed to the CalmAn Online algorithm, encapsulated in a custom actor, that calculated neural spatial masks (ROIs), fluorescence traces, and estimated (deconvolved) spikes across time for each neuron, shown for three example traces. **f** A linear-nonlinear-Poisson (LNP)

model was reformulated to work in the streaming, one-frame-at-a-time, setting. *Center*, Diagram of our model incorporating stimuli, self-history, and weighted connection terms. *Bottom*, Log-likelihood fit over time, as more frames were fed into *improv*. The online model fit converged to the offline fit obtained using the full data set (dotted line) after a single repetition of unique visual stimuli (shaded region). **g** The 'Data Visualization' actor was a GUI that displayed the raw acquired images (*left*), a processed image overlaid with ROIs color-coded for directional preference, neural traces for one selected neuron (white), and population average (red), tuning curves, and model metrics. The processed frame showed each neuron colored by its directional selectivity (e.g., green hues indicate forward motion preference). The LNP actor interactively estimated the strongest functional connections to a selected neuron (green lines). The LNP model likelihood function (bottom right) showed the optimization loss across time and estimated connectivity weights of highly connected neurons below.

Supplementary Video 2). Thus, real-time modeling with *improv* allowed for rapid identification of brain-behavior relationships that could ultimately be used to time causal perturbations to the system based on current neural or behavioral state.

### Streaming predictions of future neural activity

Given the growing importance of testing theories based on neural population dynamics[10,11,52], we next asked whether *improv* could be used to learn and predict upcoming neural population activity during a single experiment. As neural dynamics, by definition, vary across time, and are highly individual- and trial-specific, it is important to learn and

track these trajectories as they occur. By doing so, experiments could directly test hypotheses about how these neural dynamics evolve across time by perturbing activity at precise moments along neural trajectories. In this simulated example, we tackled the first stage of such an experiment by modeling latent neural dynamics in real time and generating future predictions of how trajectories would evolve given current estimates of the neural population state. Here, we used data[53] recorded from the primary motor cortex (M1) in an experiment where monkeys made a series of self-paced reaches to targets.

For this pipeline, note that we did not need to reimplement or change the code for the proSVD actor from our previous experiment

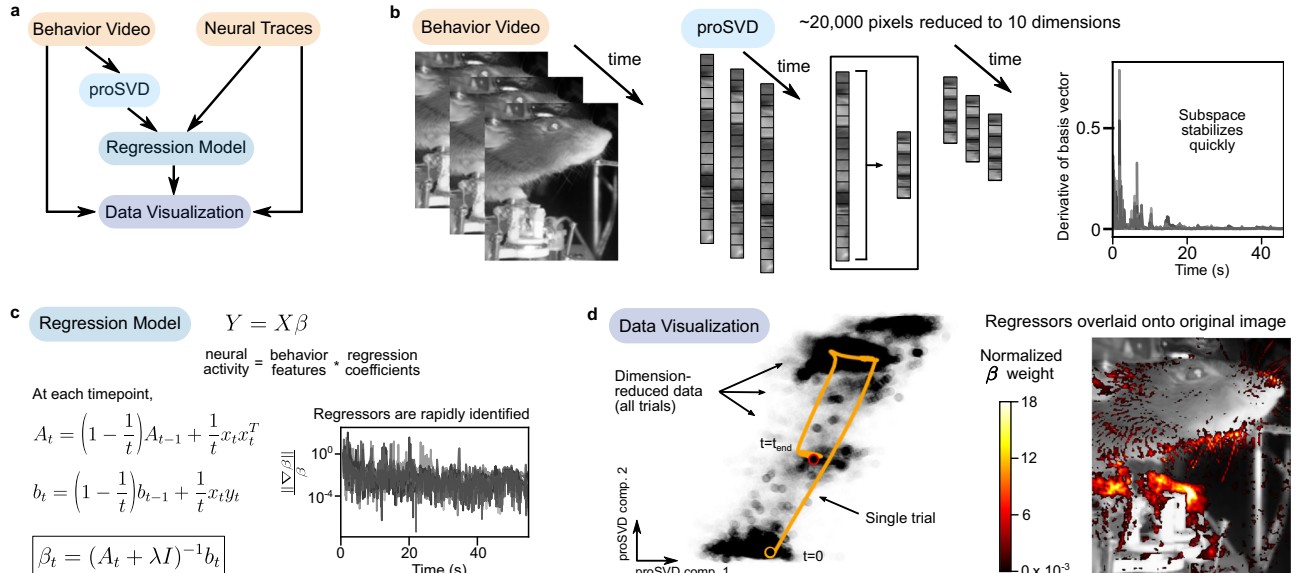

**Fig. 3 | _improv_ handles concurrent neural activity and behavioral video data streams. a** _improv_ pipeline for processing behavioral video and neural activity traces, implementing streaming dimension reduction, streaming ridge regression, and real-time visualization. **b** Video frames were streamed from disk at the original data rate of 30 frames/sec. After downsampling images, a 'proSVD' actor implemented the dimension reduction algorithm. The learned 10-dimensional proSVD basis vectors stabilized within less than a minute. **c** The 'Regression Model' actor received the dimension-reduced video data and neural activity traces and computed the regression coefficients β. For model fitting, ridge regression was implemented via a streaming update algorithm in which each datum is only seen once.

Here, _Y_ represents the matrix of neural data (57 neurons x time) and _X_ represents the matrix of reduced behavioral data (10 proSVD dimensions x time). Different gray lines correspond to different coefficients for each latent behavioral feature (10 total). **d** Two data visualization methods were used for monitoring during a simulated experiment. _Left_, Video data (dots) were plotted in the proSVD space with a representative trial highlighted in orange. _Right_, Regression coefficients were normalized to the top coefficient and overlaid back onto the original behavioral image by projecting from the proSVD basis. Regions of the mouse's face and paws are most predictive of the simultaneously occurring neural activity (_cf._, Musall[49], Fig. 3h).

(Fig. 4a). Rather, we easily inserted this module into a new pipeline simply by modifying a parameter file with dataset-relevant values describing the neural data (Fig. 4b). With _improv_, it is thus extremely simple to combine old and new analyses by reusing actors or swapping models – either in new experiments or during an experiment in progress. As our new experiment, we thus acquired neural data in the form of sorted spikes, used proSVD for streaming dimension reduction on the neural data, implemented another streaming algorithm to learn and predict latent neural trajectories, and visualized the model metrics and projected neural paths.

After dimension reduction on the neural data, we used a streaming probabilistic flow model, Bubblewrap[50], to model the resulting low-dimensional latent trajectories in real-time (Fig. 4c). By covering the observed latent trajectories with Gaussian tiles (or 'bubbles'), the model maximized the likelihood of observed transitions between these tiles, learning a transition matrix _A_ that allowed it to predict the likely evolution of the current trajectory into the future. Predictions even one full second (100 samples) into the future remained accurate, dropping in performance by only 11% from one-step-ahead predictions, as quantified by the log predictive probability (Fig. 4d). Thus, this model can, in principle, be used to plan causal interventions of these neural trajectories and precisely time their delivery. Importantly, such trajectories or perturbations cannot be known in advance, and thus, real-time predictions of ongoing neural activity are essential for conducting true causal tests of neural population dynamics theories.

**Closed-loop stimulus optimization to maximize neural responses**

Causally perturbing neural dynamics in adaptive experiments requires real-time modeling to efficiently determine when, where, and what kind of feedback to deliver. Instead of being restricted to a pre-defined and limited set of options, we instead used _improv_ during live, two-photon calcium imaging to choose visual stimuli based on ongoing

responses to visually evoked neural activity in larval zebrafish (Fig. 5, Supplementary Fig. 6). A standard experimental design is to present a limited set of motion stimuli while simultaneously imaging neural activity to measure each neuron's direction selectivity[20,54]. However, because of time constraints set by the number of stimuli and presentation duration, it is difficult to assess a larger stimulus space if sampling is not optimized. For instance, previously, we needed about eight minutes per plane (20 s per stimulus * 3 repetitions * 8 directions) for such a coarse directional tuning curve (45° intervals)[20]. Yet, evaluating each neuron for its response to twenty-four different angles (at 15° intervals) for each eye results in 576 possible stimuli (20 s * 24 per left eye * 24 per right eye * 3 reps) that would take close to ten hours for a single plane alone or over 200 h of continuous imaging for the entire brain.

Here, we implemented an adaptive approach using Bayesian optimization (BO)[55] to quickly determine fine directional tuning. Avoiding complex software-hardware integration, we applied ZMQ libraries to rapidly transfer fluorescence images via an Ethernet connection and communicate with _improv_, controlling stimulus parameters on the fly ("Methods"). _improv_ utilized a 'Bayesian Optimization' actor (BO) to select which visual stimulus to display on each next trial to maximize a given neuron's response to visual stimuli. To initialize the BO model, the responses to an initial set of eight stimuli were analyzed by the 'Caiman Online' actor (Fig. 5b). A Gaussian Process (GP)[56] was then used to estimate a given neuron's tuning curve _f_ across all stimulus possibilities, as well as the uncertainty σ in that estimate (Fig. 5c). We then chose the optimal next stimulus based on a weighted sum of those two components, balancing exploration and exploitation[57]. This cycle of acquiring and analyzing the neural responses, updating the model estimate, and selecting the next stimulus continued until a chosen model confidence value or an upper limit on the number of stimuli ($n_{max}=30$) was reached (Supplementary

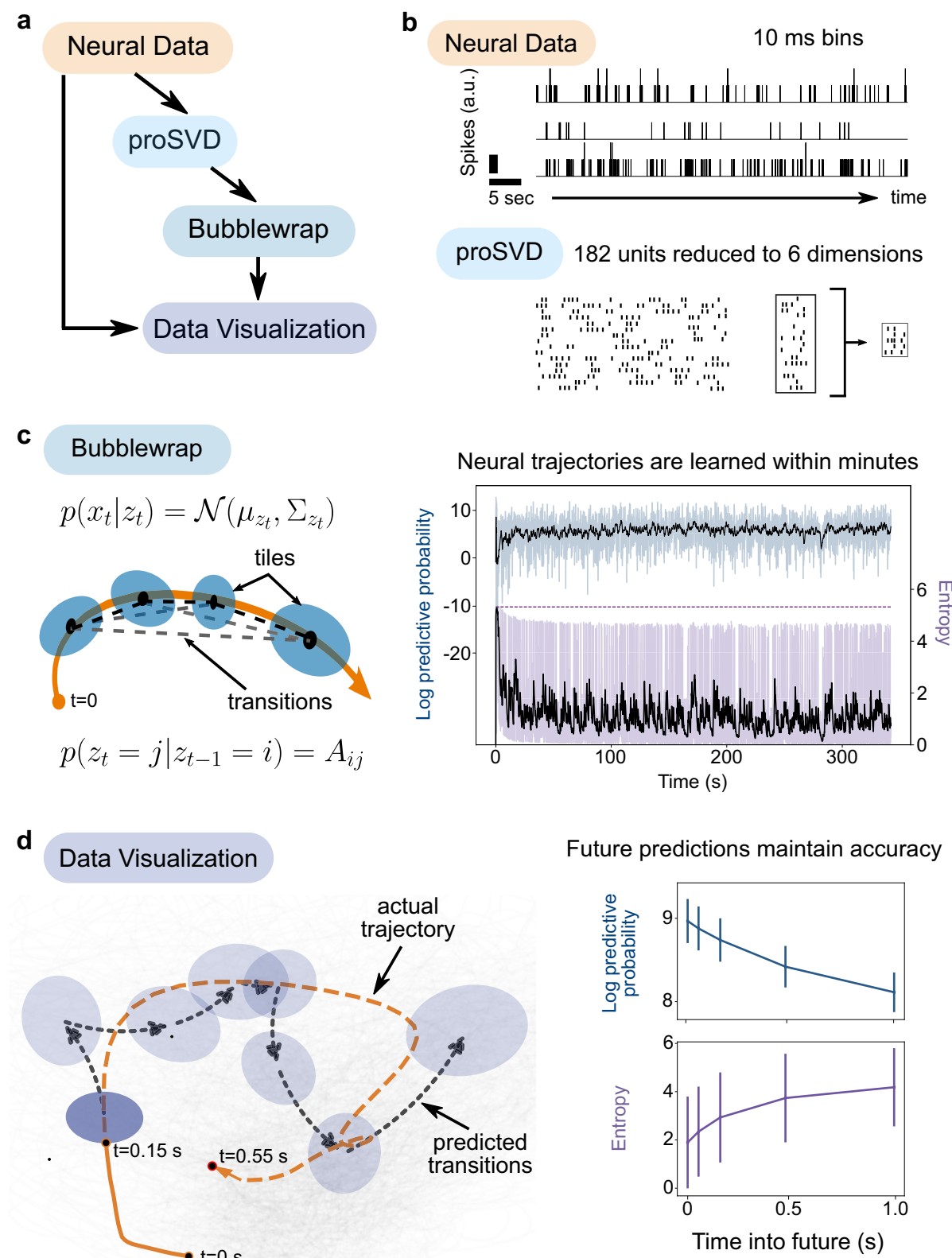

Video 3). Next, a new neuron was randomly selected from all respon-
sive neurons for optimization using the same procedure.

We validated our approach by comparing it to rigid sampling
within a reduced stimulus space (144 unique combinations) and found
that our online BO approach identified qualitatively similar neural
response curves compared to the offline GP fit to all collected data
(Fig. 5d). In addition, we quantified the accuracy of the identified peak
by computing the Euclidean distance between the location of the
maximum values of the offline and online GP fits, accounting for cir-
cular boundary conditions (Supplementary Fig. 7). On average, the
correct peak was identified in 93% of neurons chosen for optimization,
and incorrectly identified peaks tended to have more complex tuning
curves with multiple maxima (Supplementary Fig. 8). Better accuracy
could be achieved by increasing the desired confidence level.

**Fig. 4 | Real-time latent neural trajectory prediction with *improv*. a** *improv* pipeline for dimension-reducing multichannel neural electrophysiology data and predicting latent dynamics in real-time. Data from (O'Doherty et al.[53]). **b** Neural spiking data are streamed from disk, simulating online waveform template matching, binned at 10 ms, and smoothed using a Gaussian kernel to obtain firing rates. The 'proSVD' actor then reduced 182 units down to a stable 6-dimensional space. **c** The 'Bubblewrap' actor incorporates dimension-reduced neural trajectories and fits (via a streaming EM algorithm) a Gaussian mixture Hidden Markov Model to coarsely tile the neural space. *Left*, A dimension-reduced input data trajectory (orange line), bubbles (shaded blue ellipses), and the (probabilistic) connections between bubbles (dashed black line). *Right*, The model predictive

performance is quantified by the log predictive probability (blue, top) and the entropy of the learned transition matrix (purple, bottom). Black lines are exponentially weighted moving averages. **d** Predictions can be qualitatively and quantitatively monitored via *improv*. *Left*, Dimension-reduced neural data are displayed in light gray with the neural trajectory of the current arm reach shown in orange; bubbles and connections as in **c**. The dashed black line indicates the predicted transitions in the space given the first 150 ms of the trial, predicting 400 ms into the future. *Right*, Bubblewrap's predictive performance (log predictive probability and entropy; mean and standard deviation) is shown as a function of seconds predicted ahead. Error bars denoting standard deviation are calculated across all timepoints in the second half of the dataset ($n = 7500$).

While this method only optimized stimuli to drive peak responses of a single neuron at a time, simultaneous imaging of the entire population allowed us to observe the responses of all other neurons in the field of view, and thus we updated tuning information for all neurons after each stimulus. This meant that, in practice, just 15 stimuli were needed (on average) to optimize a population of 300 neurons (Fig. 5e). Thus, using *improv*, we quickly identified the peak tunings across different neural populations (Fig. 5f). For instance, when comparing peak tunings for neurons in the pretectum (Pt) and the optic tectum (OT), we observed differences between regions, differences that were reflected in the algorithm's pattern of stimulus sampling. We noted that Pt neurons preferred whole-field stimuli where the same angle was shown to each eye, but OT neurons' peak tunings were concentrated in off-diagonal regions, with converging or diverging stimuli displayed to the fish. The adaptive algorithm correctly chose to sample in regions where neurons were maximally tuned, depending on the region (Fig. 5f, white 'x's). Again, since we record simultaneously from all neurons, each new optimization routine leveraged data from the previous neurons, effectively aggregating information across the given neural population. Thus, this method is particularly well suited for applications where population correlations are expected, allowing for exploring larger stimulus spaces.

## Adaptive optogenetic photostimulation of direction-selective neurons

Finally, we used *improv* to adaptively select neurons for optogenetic photostimulation based on direction selectivity. Optogenetics is powerful for dissecting causal neural interactions, activating neurons by opening light-gated channels while recording from downstream target neurons[16,58]. Yet, typically, the criteria for photostimulation, including location and neuron type, must be learned beforehand. Here, we leveraged *improv* to implement real-time data analysis to enable more flexible phototargeting that cannot be pre-specified, such as the functional roles of individual neurons (Fig. 6; Supplementary Fig. 6). Specifically, we used an all-optical approach[59] in larval zebrafish, simultaneously performing two-photon photostimulation in red (1045 nm) of neurons expressing a novel redshifted marine opsin, rsChRmine[60] during two-photon calcium imaging in green (920 nm, GCaMP6s), avoiding spectral overlap.

After alignment of the imaging and photostimulation paths before each experiment (Methods; Supplementary Fig. 9), *improv* coordinated phases of rapid characterization of individual and network-level calcium responses, followed by a closed-loop, automated photostimulation of neurons identified by their visual response profiles (Fig. 6b, d). We designed this adaptive procedure to probe how functionally-grouped and direction-selective neurons might drive the visually-responsive neural circuits in ways similar to the presentation of external stimuli. First, we characterized individual neurons' responses to a predetermined set of motion stimuli (Fig. 6c). Next, these tuning curves were used by our 'Adaptive Targeting' actor to select a neuron with specific direction preference and opsin expression level for photostimulation. The targeted neuron's x and y position estimated from the 'Caiman Online' actor was then sent to the

'Photostimulation' actor for immediate photostimulation (Supplementary Video 4). *improv* orchestrated the automated analysis of 3–5 repetitions of photostimulation-evoked responses in neurons across the entire field of view. This targeting and stimulation procedure was repeated by selecting new neurons based on continually estimated criteria, e.g., directional tuning, excluding previously stimulated neurons, and opsin expression. Therefore, *improv* enabled real-time tracking and manipulation of experimental parameters to phototarget relevant neurons as they were discovered during imaging throughout the experiment.

Using these methods, we identified neurons that responded to photostimulation of a specific target neuron (Fig. 6e). We observed a variety of photostimulation-related response profiles, with some neurons exhibiting no response to any photostimulation, some with selective activation (Fig. 6f, 1&2), and some with consistent photostimulation-locked responses independent of the targeted neuron (Fig. 6f, 3). When combined with information about each neuron's responses to directional visual stimuli, these data allowed us to examine the ways in which different directionally tuned neurons interact. Across fish, we characterized individual neurons for their own directional preference, as well as their responsiveness during the photostimulation of other neurons (Fig. 6g). We observed some neurons whose visual tuning curves closely match their photostimulation tuning curve, suggesting that their visual response may be at least partly driven by neurons with similar tuning properties. For example, some forward-selective neurons were activated by photostimulation of other forward-selective neurons. Interestingly, some neurons responded more during photostimulation than to visual stimulation, while others stopped responding when neurons were photostimulated with shared direction preferences. Overall, slightly more neurons were more responsive to visual stimulation, often responding to both visual stimuli and photostimulation (Fig. 6h, right). In general, neurons that were more activated by photostimulation were either weakly activated by visual stimulation or responsive only to photostimulation events.

Together, these experiments demonstrate *improv*'s ability to orchestrate a complex adaptive experiment involving stimulus selection and evaluation, experimental phase switching, data handling, and online target selection, including conversion of coordinates of the imaging path into voltage signals for photostimulation. *improv* enabled us to compute diverse responses to visual and photostimulation in real-time and to dynamically select neurons for perturbation based on ongoing estimates of their network-driven activity. Future experiments could extend these paradigms to combine both visual stimuli and photostimulation simultaneously, incorporating higher-dimensional stimuli via Bayesian optimization, considering behavior, or rapidly dissecting functional connectivity across brain regions.

## Discussion

As data collection capacities in neuroscience continue to grow, the key question for experimentalists may no longer be which data they can afford to collect but which they can afford to ignore. Adaptive experiments like those illustrated above offer a powerful alternative in

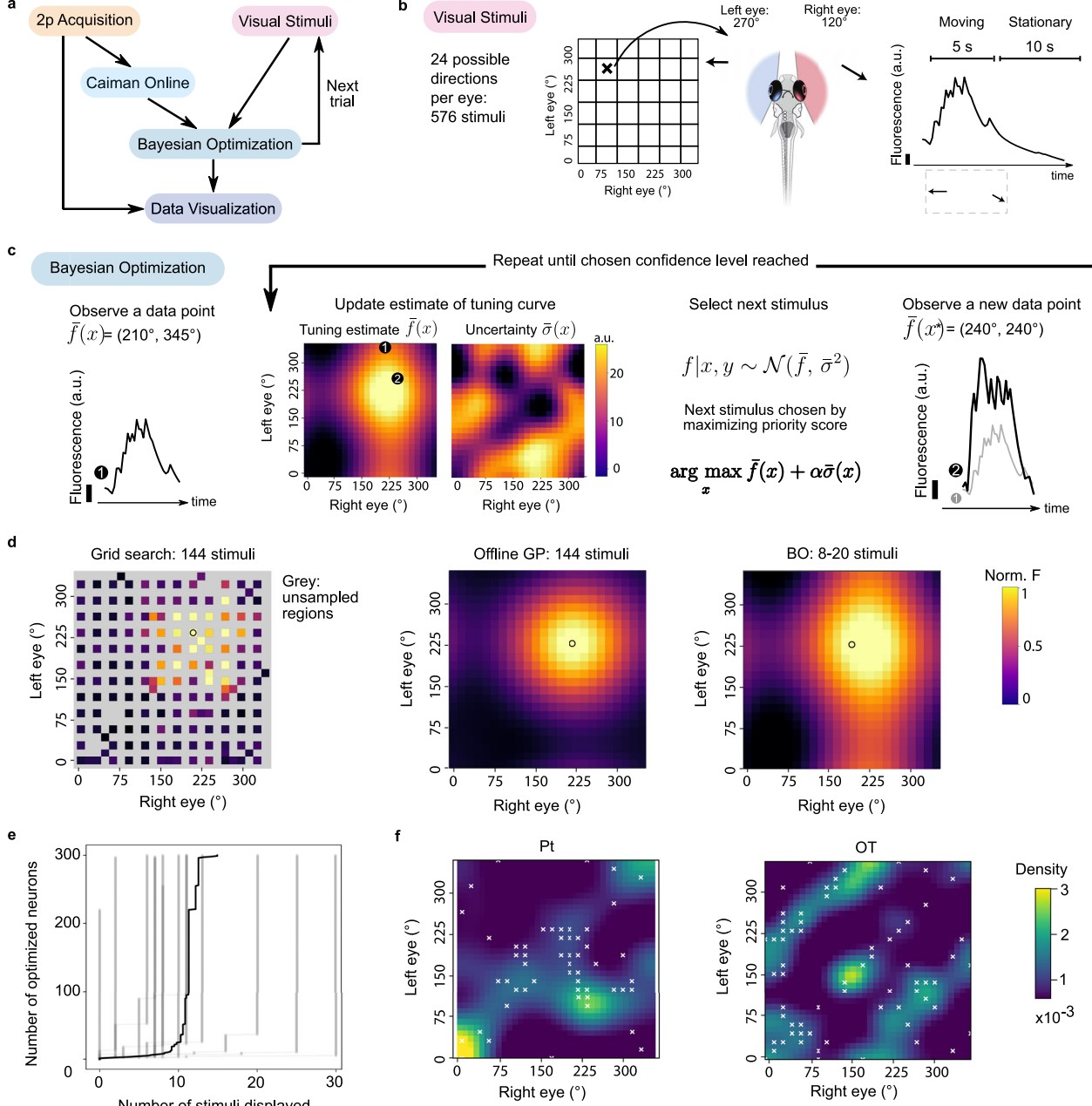

**Fig. 5 | *improv* enables closed-loop optimization of peak neural responses.**
**a** *improv* pipeline for optimization of neural responses during calcium imaging using a 'Bayesian Optimization' (BO) model actor to inform the 'Visual Stimuli' actor to select the next stimulus to display. **b** Matrix of 576 possible combinations of visual stimuli consisting of moving gratings shown to each eye individually. Stimuli are moving for 5 s and held stationary for 10 s. **c** Online BO actor assesses a neuron's stimulus response to the current visual stimulus and updates its estimate of the tuning curve using a Gaussian process (GP) (*f*), as well as the associated uncertainty (*o*). The next stimulus is selected by maximizing a priority score that balances exploration (regions of high uncertainty) and exploitation (regions of high response). Estimates and uncertainty are plotted here and in (**d**) using a color scale normalized to 1 for visualization. **d** To measure similar receptive field precision, the online BO approach typically only requires 8–20 stimulus presentations, compared to an incomplete grid search of 144 stimuli (gray denotes unsampled regions). The peak tuning identified by an offline GP fit and the empirical peak tuning from the grid search agree with the peak tuning determined online. **e** On average, just 15 stimuli are needed to determine peak tunings of 300 neurons in real time ($N = 12$ imaging sessions). **f** Heatmaps showing the distributions of identified peak tunings for individual neurons in the pretectum (Pt, *left*) or the optic tectum (OT, *right*). Color indicates the density of tuning curve peaks across the population. White 'x's mark the locations where the algorithm chose to sample. In this example, the algorithm sampled primarily near the diagonal (congruent, same direction of motion to both eyes) in the Pt but chose to sample more frequently in off-diagonal areas (different direction of motion to both eyes, e.g., converging motion) in the OT.

which models guide data collection and causal interventions based on continuous feedback. *improv* dramatically simplifies the design and prototyping of these experiments by combining flexible, modular specification of analysis pipelines with arbitrary modeling in the loop. *improv* accomplishes these complex tasks while orchestrating the most time-consuming and tedious aspects of constructing adaptive pipelines: parallelism, error-handling, and data management flows (Fig. 1). With *improv*'s software architecture, we show that many commonly used models, such as linear-nonlinear Poisson and regularized regression, can easily be fit online, often requiring no more

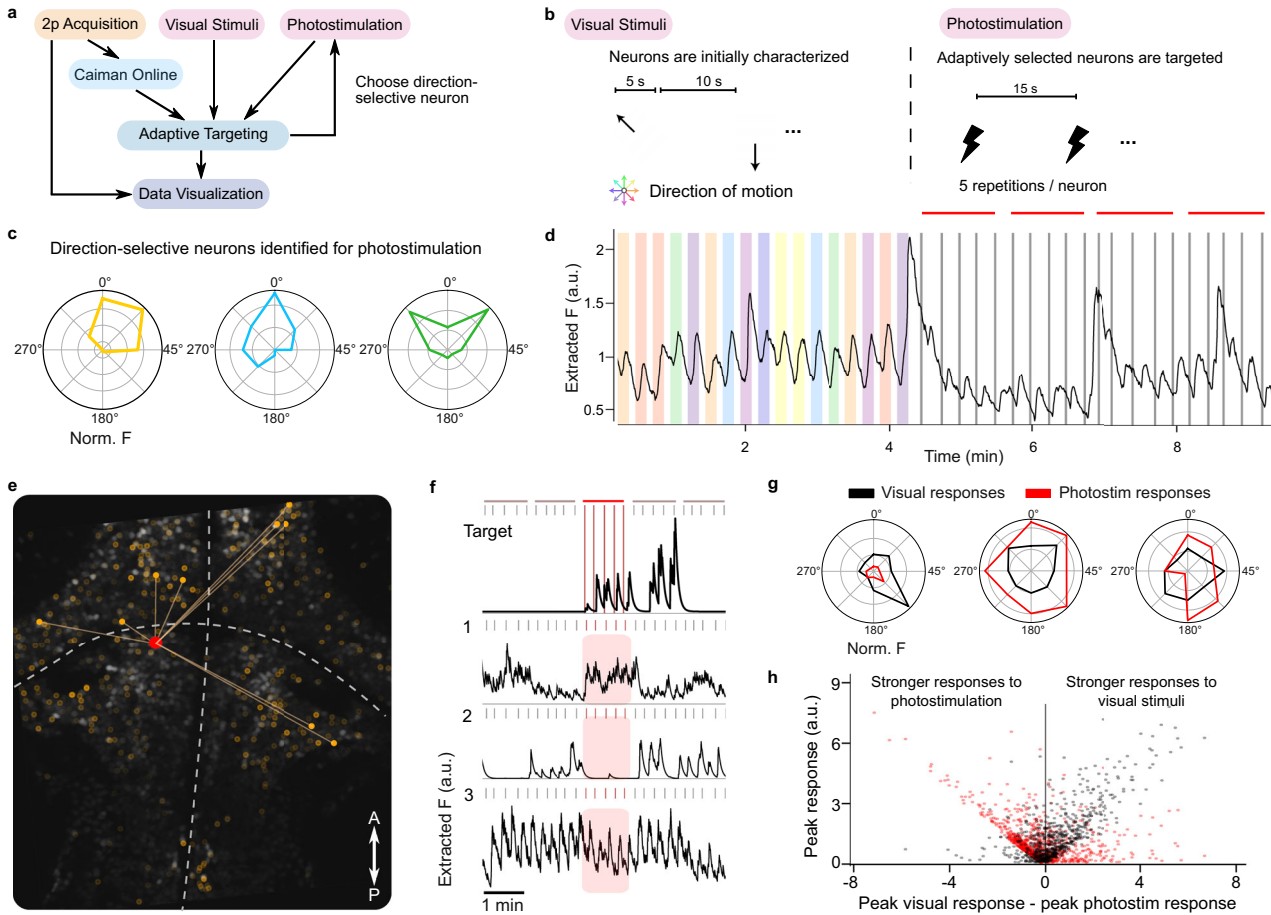

**Fig. 6 | Adaptive optogenetic photostimulation target selection during functional calcium imaging in zebrafish. a** *improv* pipeline for targeting direction-selective neurons for photostimulation. The 'Adaptive Targeting' actor provides the target locations to the 'Photostimulation' actor to control the next photostimulation event. **b** *improv* orchestrates a phase of visual stimulation to characterize neurons in real time, automatically switching after reaching characterization criteria to a closed-loop photostimulation phase. **c** Example visual tuning curves of neurons chosen for photostimulation, normalized to their maximum response. The 'Adaptive Targeting' actor selects a direction-selective neuron with rsChRmine expression for photostimulation. **d** Average fluorescence across all neurons showing stimulus-locked responses to visual stimuli (colored bars) followed by a photostimulation phase (gray lines). **e** Pretectum (Pt) and hindbrain (Hb) in a zebrafish expressing nuclear-targeted GCaMP6s (greyscale). Photostimulating a target neuron (red dot) results in evoked activity in responsive neurons (orange dots). Color intensity encodes the response magnitude, and lines denote the top 10 strongest responders. **f** Fluorescence responses to photostimulation of the red

neuron in (**e**) (top trace; Target), and three simultaneously recorded putatively responsive neurons (bottom): (1) responsive, upregulated; (2) non-responsive; and (3) consistently responding. Extracted fluorescence traces are obtained from 'Caiman Online'. **g** Example neurons' visual and photostimulation tuning curves. The visual tuning curve is calculated as the average response during a given angular direction of the stimulus (45°, 90°, …). The photostimulation tuning curve is calculated as the average response during photostimulation of other neurons whose tuning peaks were located at a given direction, such as the 45°, forward-right tuned neuron in **c**). The left plot shows a neuron that is less activated by photostimulation but with similar directional tuning, and the two left neurons show stronger universal activation from stimulated neurons. **h** Analysis of all neurons across fish ($N = 3$; ~1300 neurons) responsive to visual stimuli (red dot) and photostimulation events (black dot), compared to their overall responsiveness. Most neurons responded to visual and optogenetic stimulation, but some neurons were only discovered as responsive during photostimulation. Diagonal trends on either side indicate neurons preferring either visual or photostimulation.

than a few minutes to approximate their offline estimates (Fig. 2). The result is a system in which new experiments can be quickly prototyped from interchangeable parts, in keeping with the Unix philosophy[61].

In addition, to enable integration with other widely used tools, we designed *improv* to easily interoperate with many packages not designed for streaming settings or real-time execution. For example, Suite2p[6] can be used as a batch preprocessor for groups of images as a drop-in replacement for CaImAn (Supplementary Fig. 10) and this functionality can be implemented in just a few lines of code. Currently, *improv* is written in Python, which we chose for the language's wide appeal and native support of many machine learning libraries, but as proof of concept, we implemented some analysis algorithms in the Julia language[62], which can offer better performance through just-in-time compilation. While we constructed GUIs to operate most of the simulated and in vivo experiments we ran here, *improv* can run on low-resource systems using only a text-based command line interface,

allowing users to customize the system based on their needs (Supplementary Video 5).

It is this flexibility that distinguishes *improv* from similar software systems. For instance, BRAND, which likewise focuses on real-time experiments and shares many architectural features with *improv*, runs only on Linux, with core code written in C++[63]. Similarly, Heron which uses a graph structure to specify experimental flow and offers a GUI for interactive design of experiments, does not focus on performance and runs only on Windows[64]. Bonsai, which constructs experiments using a visual programming interface or custom C# scripts on Windows, is useful for simpler experiments but is less suitable for complex workflows or integration with modern machine learning methods that use Python[65]. Cleo, which targets closed-loop experiments, is focused on detailed biophysical simulation of the systems involved[38]. Moreover, while specialized systems like that of[33] have also established closed-loop setups in the zebrafish, these methods rely on specialized

hardware and software solutions that are tightly matched to the specific system under construction. *improv*, by contrast, offers a cross-platform, modular, domain-agnostic system that maximizes the flexibility available to users when incorporating multiple data streams and models into real-time experiments.

This flexibility, along with a commitment to cross-platform support and use of commodity hardware, also necessitates tradeoffs in *improv*'s function in the real-time setting. Only the most recently acquired data is relevant to maintain in memory, and thus *improv* ignores old data to optimize speed. Typically, only data from the last few images or time bins are kept in memory for active use by streaming algorithms before being offloaded to hard drive storage. Our use of an in-memory data store, which gives all actors fast and easy access to all data, also requires a fairly large amount of dedicated memory, which limits what else could be run on the same computer. Still, all *improv* applications always stayed ahead of data acquisition, applying modifications to some algorithms: for example, when using CaImAn Online, we ceased updating the shapes of neurons after an extended imaging period (> 30 min) and stopped adding new neurons to consider once a threshold was reached.

Another key limitation inherent to all real-time systems is the necessity of synchronizing multiple pieces of hardware and multiple data sources. Despite *improv* naturally accommodating actors spread across different machines, it currently only supports a single, centralized data store and orchestrating server, each of which run on a particular machine. While this architecture represents a potential limitation, it is one shared by all other real-time systems[33,63,64] and posed no practical challenge in any of our in silico or in vivo experiments.

As both the volume and variety of data collected in neuroscience continues to grow, so will the need for new experimental designs that can cut through this complexity to efficiently test hypotheses[19]. We have argued that adaptive designs, particularly those that incorporate models into the process of data collection, offer a way forward. *improv* provides a tool for engineering adaptive designs and thus opens the door to new classes of experiments in which large hypothesis classes can be effectively tested by data-driven algorithms, bridging large-scale experiments and complex post hoc analyses. By simplifying the inclusion of streaming analysis and visualization into data collection, *improv* allows experimenters to gain real-time insights as experiments progress, allowing them to explore and refine their ideas during the pilot phase. Thus, we expect our platform to provide an empowering set of tools for integrating data collection with analysis and intervention across a broad range of experiments in neuroscience.

# Methods

## Datasets
**Zebrafish functional types and connectivity.** Previously recorded raw two-photon imaging data containing visual stimulus information and two-photon calcium images from a single plane in the pretectum were used in Fig. 2[20]. Images in a custom format (.tbif) were streamed from disk at the rate of earlier data acquisition, 3.6 Hz, into the data store using a custom 'Tbif Acquirer' actor.

**Mouse behavior and neural activity.** The mouse behavior video with two-photon calcium fluorescence traces dataset used in Fig. 3 can be found at https://doi.org/10.14224/1.38599, and were originally presented in ref. 49. We used the dataset mSM49 recorded on July 30, 2018, and the Camera 1 recording for the behavior video.

**Monkey neural spiking activity for trajectory prediction.** The electrophysiology dataset from a monkey reaching task used in Fig. 4 can be found at https://doi.org/10.5281/zenodo.3854034. Note the data were not separated into trials, allowing for continuous trajectory estimation. We used the indy_20160407_02.mat dataset and binned at

10 ms. We also applied a smoothing Gaussian kernel with a 50 time-point window length to obtain a continuous estimate of firing rates for each channel.

## Experimental integration
Our example implementation on *improv* and its integration with live experiments has been tested on three main computers: (1) a 2015 Macbook laptop with 8 GB of RAM and a 2 core 2.2 GHz Intel i7 processor; (2) a 2018 custom-built (total cost <$4k) Ubuntu desktop machine with 128 GB of RAM and a 14 core 3.1 GHz Intel i9 processor; and (3) a 2019 custom-built (< $4k) Windows desktop machine with the same specifications as (2). Our package has been confirmed to be operational with all major operating systems (Windows 10 with Windows Subsystem for Linux, Ubuntu, and Mac OS X).

The integration of *improv* into the experimental two-photon microscopy setup required only a method for transferring streaming data from the acquisition system to our analysis system in real time. We ran *improv* on a separate computer, but in general a separate computer is not required, and analysis can be done on the same machine. While many networking solutions exist, we chose to use ZeroMQ, a widely used universal messaging library that has interfaces in many languages, including Python and LabVIEW. Images obtained in LabVIEW were thus directly streamed via ZeroMQ to our 'ZMQ Acquirer' actor in Python. Similarly, messages and timestamps for the visual stimulus being displayed were also streamed directly from the Python script running the visual stimuli to the same actor. And with *improv*'s flexibility, should we want to run a simulated experiment using data streamed from disk (rather than live via ZeroMQ), only the Acquirer actor would need to be changed, leaving the rest of the pipeline intact.

For the experiments reported in Figs. 5, 6, we also generated some lab-specific integration code that is hosted at: https://github.com/Naumann-Lab/improv.

## Visual stimuli
Stimuli were presented from below using custom Python 3.10 software (*pandastim*: www.github.com/Naumann-Lab/pandastim). Using a 60 Hz, P300 AAXA Pico Projector, stimuli were presented using only the red LED. This created red and black square-wave grating patterns, providing visual stimuli that strongly evoke optomotor response behaviors[66]. Stimuli were created online from dual dictionaries, determining the texture and the stimulus object. Textures were created as dual gratings presented independently to each eye of the zebrafish and provided parameters for the dark value, the light value, the spatial frequency, and the texture size. These textures were moved in accordance with the parameters specified in the stimulus object. The parameters independently controlled the two stimulus textures to specify the stationary duration (time pre-movement), the moving duration, the speed of movement, and the angle of movement. The texture and stimulus parameters were parsed into a combined stimulus class and held in an ordered queue for serial display. Adaptive visual stimulation experiments presented in Fig. 5 used a stationary time of 10 s, moving time of 5 s, speed of 10 mm/s, and different angles of movement in each eye.

## Zebrafish
All experiments with live zebrafish (*Danio rerio*) were approved by Duke University's standing committee on the use of animals in research and training. We raised zebrafish larvae in small groups of about 30 fish in filtered embryo water on a 14-h light, 10-h dark cycle at a constant 28 °C. From 4-days post fertilization (dpf) onwards, we fed larvae with paramecia. Embryos were kept in E3 solution (5 mM NaCl, 0.17 mM KCl, 0.33 mM CaCl2, 0.33 mM MgSO4). We performed all calcium imaging experiments with zebrafish aged 5–7 dpf. At these stages, sex cannot be determined. For calcium imaging alone, we used

pigmentless Casper or nacre *Tg(elavl3:H2B-GCaMP6s)*[44,67], screened for homozygous, bright fluorescence expression. These fish were supplied as a generous gift from Dr. Misha Ahrens[26].

For optogenetic photostimulation experiments, we generated a new transgenic line expressing a red-shifted marine opsin, rsChRmine[60] under the elavl3 promoter. *Tg(elavl3:rsChRmine-oScarlet-Kv2.1)* were generated by standard Tol2 mediated transgenesis, resulting in strong neural expression visualized as bright, red fluorescence emitted by the red fluorescent protein oScarlet. The somatic potassium channel Kv2.1 motif was added to boost the expression of rsChRmine in each neuron's soma area rather than in neural processes. These fish were outcrossed to generate the double transgenic fish line *Tg(elavl3:H2B-GCaMP6s); Tg(elavl3:rsChRmine-oScarlet-Kv2.1)*. These zebrafish were screened around 36 h after fertilization for red and green fluorescence in the brain and spinal cord, as well as strong light-induced movement, indicating functionality of the opsin.

## Two-photon imaging

In vivo two-photon fluorescence imaging was performed using a custom-built two-photon laser-scanning microscope, equipped with a pulsed Ti-sapphire laser tuned to 920 nm (InsightX3, Spectraphysics, USA). To minimize movement artifacts, larval zebrafish were embedded in low-melting-point agarose (2% w/v), with their tails freed to allow for observation of behavioral responses. In addition, viability monitoring was supplemented by observing the heartbeat and blood flow through brain vasculature before and after imaging. All data acquisition was performed using custom LabVIEW (National Instruments, USA) and open-source Python code. Typically, the images were obtained in raster scanning mode with 512 × 512 pixels scanned at 400 kHz, and frames were acquired at 2.3 Hz, with imaging power of ~10 mW at the specimen.

To account for possible vertical drift of the fish over time, we ran a stabilization program that performed an automatic alignment check every 10 min, pausing visual stimulation for the duration of the alignment check. Our alignment check involved taking a small image stack around our current position, using 5 imaging stacks. This stack began one neuron depth below our current plane (−6 μm) and progressed to one neuron depth above our current plane (+6 μm), generating images at (−6, −3, 0, +3, +6) μm away from our current imaging plane. The images were binarized using an Otsu threshold[68], and then compared to the target image acquired during experiment initialization via the Sørensen–Dice coefficient (DSC)[69]:

$$DSC = \frac{2|X \cap Y|}{|X| + |Y|} \quad (1)$$

From this, the plane was automatically moved to the highest scoring plane. This alignment method allows us to continuously image across a stable plane for durations of over 3 h.

## Two-photon calcium fluorescence analysis

Spatial and temporal traces were extracted from the calcium images using the CaImAn Online algorithm within CaImAn[7], modified with custom code to allow for single frame-by-frame processing of incoming data streams without prespecifying the data length. Parameters supplied to CaImAn Online for motion correction and source extraction are specified in a configuration file to the preprocessing actor and are available in our codebase. Our analyses utilized both the estimated fluorescence traces as well as the extracted spike counts for subsequent model fitting. Extracted spatial traces were used for visualization in the user interface with openCV to fill in and color each neuron by its responses to visual stimuli.

To compute each neuron's direction selectivity, a running mean was calculated for responses to each of the 8 directions of visual motion. For baseline subtraction, we averaged a window of 10 frames

directly before the onset of each motion stimulus. We averaged 15 frames during motion to calculate the neuron's response to each motion stimulus. The relative response magnitudes to each stimulus were calculated to color encode response magnitude (brightness) and directionality (hue), resulting in a visual representation of anatomical direction selectivity distribution (Fig. 2, see arrow wheel for color code).

## LNP model fitting

In zebrafish calcium imaging experiments (Fig. 2), we used a form of the well-known linear-nonlinear-Poisson model to estimate functional connections among all observed neurons[47]. Our specific version of the model included terms for: (1) a baseline firing rate; (2) the stimulus responses, simply modeled as a vector of length 8 corresponding to the current stimuli, without using any spike-triggered averages; (3) the self-history effect, modeled using a history vector of activity up to 4 frames prior to the current frame; and (4) weighted functional connections to all other neurons, modeled using the prior frame activity information. An exponential nonlinear term was used to compute firing rates.

To fit this model online, we used stochastic gradient descent with windows of data ranging from 10 to 100 frames of prior data held in memory. Step sizes were chosen based on the data, but a value of 1e-5 was generally used successfully. For visualization purposes, each neuron's top 10 connections (determined by magnitude) were sorted and displayed in the graphical user interface as both a matrix (right) and green lines (center, if a neuron is selected).

## proSVD and ridge regression

For streaming dimension reduction of the behavioral video, we used the proSVD algorithm described in ref. 50. For each frame of the video, the image was first downsampled by a factor of 4 (from 320 × 240 to 160 × 120 pixels). The first 10 frames of data were used to initialize proSVD and reduced to 10 dimensions. Each subsequent frame was embedded 1 frame at a time to a stable 6-dimensional space. These projected data were then used in a streaming ridge regression derived from[51] to predict the 57 neural traces (calcium fluorescence) from the proSVD-discovered features. Lambda was fixed at 1e-5, though we note that this hyperparameter could be re-estimated on an ongoing basis as more data are acquired.

## Real-time predictions with Bubblewrap

To demonstrate real-time neural trajectory prediction, we used the Bubblewrap algorithm described in ref. 50. We used the same 'proSVD' actor employed in the previous example: we initialized using the first 100 timepoints of data, reduced each timepoint of subsequent data to 6 dimensions, and put the resulting data projected onto that subspace into the central plasma store. We constructed a 'Bubblewrap' actor that then took this data and fit the Bubblewrap model to it. Bubblewrap was initialized with the first 20 timepoints of projected data (6-dimensional) and 50 tiles in the low-dimensional space. We chose hyperparameters based on prior work[50]: lambda and nu are 1e-3, and the gradient step was 8e-3. We computed the log predictive probability and entropy at each time point post-initialization and at 1–5 time points into the future.

## Online Bayesian optimization

We used a Bayesian optimization framework to adaptively select which visual stimulus to display during a live experiment based on ongoing estimates of neural activity. We coded a custom kernel (2-dimensional Gaussian, radial basis function) and optimizer class to fit to the 2-dimensional neural tuning curves (0–360 degrees of angles of motion for each eye in 15-degree intervals). We modeled this discrete space with unit distances based on the size of the space (0–23 for each dimension) and corresponding length scale 1/24. For each neuron, we

initialized using all data acquired up to that time point. For the first neuron, this was 8 stimuli of whole field motion (same angle in each eye) sampled randomly at 45-degree resolution. The optimizer used an upper confidence bound (UCB) acquisition function[57] to select the next stimulus for display. If this stimulus had already been sampled more than 5 times for that neuron, a random stimulus was selected instead (using a NumPy random seed of 1337). After each stimulus was displayed and the resultant neural activity analyzed, a stopping function was evaluated. We computed the expected improvement (EI) and compared it to a hyperparameter (1e-2) we chose based on experience. If the EI dropped below that threshold, we saved the online-estimated tuning curve for post hoc analysis and moved on to the next neuron. If, after a maximum number of stimuli (30) was reached, the EI dropped below the threshold, we added the neuron ID to a list of neurons to come back to later in the experiment, after more (hopefully informative) data was collected during the optimization of other neurons. Each neuron was chosen for optimization a maximum of 2 times, and a total of 300 neurons were chosen for optimization in each z-plane, with a maximum of 8 planes per fish.

### Online optogenetic photostimulation

To optogenetically photostimulate visually characterized neurons in a closed-loop configuration via *improv*, we performed simultaneous calcium imaging of GCaMP6s (green, 920 nm) as described above with a first imaging optical path to raster scan across a plane of the zebrafish brain and a second photostimulation optical path equipped with a lens-conjugated galvanometer scanner to excite neurons expressing rsChRmine (red, 1045 nm). The imaging and photostimulation paths were aligned vertically in z and mapped onto each other in the horizontal plane, so that the second scanner could transform the spatial coordinates provided by the first one for precise targeting. Slight shifting and rotating of these coordinates to align both paths resulted in specific voltage commands to the scanning mirrors to deliver spiral stimulation to specific neurons with the photostimulation path. We confirmed spatial x,y,z alignment by 'burning' a grid pattern into a fluorescent microscope slide (Thorlabs, FSK2) using the photostimulation path while imaging with the imaging path[70].

After alignment, the experiment started by *improv* coordinating the first phase of the experiment, that is, calcium imaging of neurons in the zebrafish pretectum and hindbrain with the 950 nm excitation imaging path. After *improv* detected sufficient characterization of the directional tuning of all neurons in the field of view, *improv* triggered a photostimulation phase. During the photostimulation phase, *improv* autonomously selected a neuron of a specific directional tuning range and red opsin expression for photostimulation, sending the x and y coordinates to the photostimulation controller. These computed x, y target coordinates are then transformed into voltage signals to create waveform signals, resulting in spirals generated by the photostimulation path galvanometer mirrors. While we continuously recorded neural activity with the imaging optical path, we then stimulated the target neurons with these photostimulation patterns. Since CaImAn source extraction does not necessarily correspond to each neuron's cell body, we increased the diameter of the photostimulation region to the typical neuronal size of 6–10 μm. For these experiments, *improv* coordinated 5 repetitions of photostimulation events spaced 15 s apart. These photostimulation spiral scanning events lasted for 200–300 ms (depending on cell size) with dwell times of 1 ms at a laser power of 10–20 mW at the sample. In a typical experiment, *improv* automatically selected approximately 25 neurons to be photostimulated. All relevant information (fluorescence images, visual stimulation information, photostimulation event times, and locations) was saved for further analysis. For the post hoc analysis, we analyzed visual response and photostimulation-derived characteristics in the form of tuning curves and computed the peak responses for each stimulus type for all responsive neurons (Fig. 6).

### Photostimulation target selection

Neurons were selected for photostimulation based on two primary factors: (1) tuning curve derived from responses to visual stimuli and (2) pixel intensity of a red-channel image above a threshold. The tuning curves and associated color arrays were calculated as described above ('Two-photon calcium fluorescence analysis'). These were used to compute an inner product with a set of desired functional types. For example, forward types can be characterized with a basis [−1, 1, −1] from the color array [red, green, blue] where red and blue are downweighted and green (forward) is preferentially selected. A set of neurons within a range (−75 < x < 75) were selected for possible consideration.

A single red-channel image was acquired at the start of the experiment at 1045 nm in the imaging path and used for all neurons. For all neurons selected first based on visual responses, the intensity of a rectangle of pixels surrounding the center (x,y) position of the neuron was summed as a proxy for opsin expression level. Next, either the neuron whose intensity was greatest or a random neuron whose intensity exceeded a threshold (50) was chosen. To increase the diversity of neurons selected for photostimulation, we kept a list of already-stimulated neurons and preferentially (but not always) selected neurons not on this list.

### Reporting summary

Further information on research design is available in the Nature Portfolio Reporting Summary linked to this article.

## Data availability

All new imaging data generated in this study have been deposited in the DANDI database under accession code https://dandiarchive.org/dandiset/001569 and can be downloaded directly. Data from previously published work can be found at locations listed in their respective citations.

## Code availability

The improv software package can be downloaded from github.com/project-improv/improv or the Python Package Index pypi.org/project/improv. It can also be found at https://doi.org/10.5281/zenodo.17079045 and https://zenodo.org/records/17079045.

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

## Acknowledgements

We thank Eric Thomson for helpful discussions and the Duke Innovation Co-Lab for technical support. We also thank Catherine Seitz, Jim Burris, and Karina Olivera for zebrafish husbandry and Misha Ahrens for generously sharing transgenic zebrafish lines. We thank Dario Ringach for complimentary access to ScanBox. A.D., M.N., C.S., D.S., M.D.L., J.M.P., and E.A.N. were supported by the NIH BRAIN Initiative R34-EB026951-01A1 and a Duke Institute for Brain Sciences (DIBS) incubator award. A.D. was supported by a Ruth K. Broad Biomedical Research Foundation Postdoctoral Fellowship Award, a Swartz Foundation Postdoctoral Fellowship for Theory in Neuroscience, and a Career Award at the Scientific Interface from the Burroughs Wellcome Fund. A.G. was supported by the Beckman Young Investigator Program. The NIH BRAIN Initiative (RF1DA056376, 1RF1-NS128895) supported P.G., E.A.N., M.N., and J.M.P. E.A.N. was supported by the Alfred P. Sloan and Whitehall Foundation. The NIH National Eye Institute (R01EY033845) supported E.A.N. and M.D.L.

## Author contributions

E.A.N. and J.M.P. conceived of this project. A.D. and J.M.P. designed *improv* with input from M.N. A.D. and J.M.P. implemented the core *improv* codebase. A.D., C.S., P.G. and D.S. wrote software components for the various use cases and interoperability with other programs. E.P. and A.G. contributed to the development of the calcium imaging analysis software for online use. M.N. and M.D.L. built the two-photon microscope and associated visual stimuli and photostimulation software. T.B. and K.D. generated the transgenic zebrafish expressing rsChRmine. A.D., M.D.L. and M.N. performed the zebrafish imaging experiments. A.D., J.M.P. and E.A.N. wrote the manuscript with input from all authors.

## Competing interests

The authors declare no competing interests.
