## [Transparent Peer Review file · Nature Communications]

improv a software platform for real-time and adaptive neuroscience experiments

Corresponding Author: Dr Eva Naumann

Version 0:

Reviewer comments:

Reviewer #1

(Remarks to the Author)

I thank the authors for this extensive revisions and clarifications. I believe this will be a useful tool for the community.

(Remarks on code availability)

I did not test it directly, but I read over the main modules and documentation.

Reviewer #2

(Remarks to the Author)

The authors have satisfactorily addressed some of the concerns raised in my review; however, not all points have been fully resolved. In particular, a central issue remains unaddressed. To properly evaluate whether this system represents a novel contribution to the field, it is essential to compare its performance against existing systems capable of addressing similar problems—such as Bonsai and Heron.

While the authors state that such a comparison is not feasible, it is worth noting that, to the reviewer's knowledge, at least for Bonsai it is possible to conduct closed-loop experiments involving online modifications of states, rules, and parameters. Without a quantitative assessment of Improv's performance relative to existing technologies in the context of at least some of the proposed use cases, it is difficult to substantiate the claim of novelty.

Regrettably, in the absence of such evidence, I am unable to recommend this work for publication in Nature Communications at this time.

(Remarks on code availability)

NA

Reviewer #3

(Remarks to the Author)

I stand by my previous comments, but the manuscript is in good shape. The authors highlight scenarios where analyses are only feasible during the experiment.

improv prioritizes ease of use and flexibility, and its design reflects this. Its reliance on an in-memory store is both its strength (it solves the coordination problem, which is indeed a big part of modern neuroscience experiments) and also its weakness (share-then-analyze is way more expensive than data reduction/analysis before message passing, especially now that in-memory stores no longer support zero-copy). It is good that the authors have committed to a maintenance strategy that does not rely on an obsolete in-memory store.

The software is open source, cross platform, and accessible, so now all that remains is to see how it will be adopted by the

community.

(Remarks on code availability)

Version 1:

Reviewer comments:

Reviewer #2

(Remarks to the Author)

I thank the authors' for the comprehensive comparison of their software with established platforms such as Bonsai and Heron. Many of the points they raise are compelling; however, some seem to arise from limited familiarity with Bonsai rather than from intrinsic shortcomings of the platform itself. I therefore strongly encourage the authors to consider the option of including the key elements of this comparison in the manuscript, enabling readers to judge for themselves whether improv constitutes the best option for their experiments or whether existing alternatives are a better fit.

(Remarks on code availability)

Shared code is in good shape.

Point by point response to reviewers' comments:

improv: A software platform for real-time and adaptive neuroscience experiments

Nature Communications revision June 9, 2025

Black – Reviewer's comments and remarks

Blue - Our response to reviewers' comments and remarks

Red - Added or changed text to the revised manuscript

Again, we thank the reviewers and editor for their helpful suggestions and feedback throughout the review process. We were pleased to see the overwhelmingly positive comments from our reviewers, suggesting that they also view software platform *improv* as a useful tool.

Given the feedback throughout, we believe that it is important to emphasize that *improv* is not just another software package for analyzing neuroscience data. Rather, at its core, this software provides a computational framework for arbitrary (streaming) analyses of incoming data as they are being collected ***during the experiment, in real time***. Our ***software design*** is as important for this goal, as it offers an easy-to-use and flexible software tool that researchers can use to collect data and immediately incorporate models-in-the-loop, thereby executing new kinds of experimental designs that are ***not possible*** otherwise or with other tools, including other impressive tools (see Bonsai below).

Again, we want to emphasize that we agree that other excellent software packages exist to enable rapid computation of particular analyses, such as CalmAn online or DeepLabCut-live and also software that allow design of useful closed loop experiments, e.g., Heron and Bonsai.

However, *improv* is the glue that allows custom analysis packages to be orchestrated together in diverse combinations to facilitate a wide variety of real-time experiments that require easy access to variables during data collection. We hope that by sharing *improv* with the wider research community, these flexible experimental configurations will become the new standard in data collection and adaptive experiments in many neuroscience labs.

In this revision, we have made no additions and changes to the manuscript but added a quantitative and qualitative comparison of *improv* to Bonsai and Heron to the rebuttal letter below.

REVIEWER COMMENTS

Reviewer #1 (Remarks to the Author):

I thank the authors for this extensive revisions and clarifications. I believe this will be a useful tool for the community.

We thank Reviewer #1 for this favorable assessment, and we also hope that this tool will be helpful for researchers who may not have considered such experiments. Our cross-platform approach, expansive test suit, and software documentation are major strengths of this project.

Reviewer #1 (Remarks on code availability):

I did not test it directly, but I read over the main modules and documentation.

We believe that our code repository is a major strength of this publication, and we are actively maintaining and updating the software.

Reviewer #2 (Remarks to the Author):

The authors have satisfactorily addressed some of the concerns raised in my review; however, not all points have been fully resolved. In particular, a central issue remains unaddressed. To properly evaluate whether this system represents a novel contribution to the field, it is essential to compare its performance against existing systems capable of addressing similar problems—such as Bonsai and Heron.

We thank the reviewer for this comment and agree that direct, quantitative comparisons would be the gold standard to demonstrate whether *improv* represents a novel contribution to the field. We completely agree that Bonsai (and Heron) are very valuable tools, with Bonsai being cited over 590 times, however *improv* presents a powerful alternative, especially for uses leveraging modern machine learning tools and complex real-time designs that deeply integrate models into live experiments.

Nonetheless, all software engineering involves tradeoffs, and for the use cases we envision for *improv*, these alternatives entail significant problems with complexity, scalability, operating system and platform limitations, and hardware compatibility challenges:

- 1) Bonsai becomes more challenging when you need to increase the complexity of experimental or data workflows beyond what is already provided. These challenges were recognized and led to the development of Heron (see Developing Heron, the progeny of Bonsai and ROS | Sainsbury Wellcome Centre).
- 2) Bonsai is written in C# and requires C# code for customization. This is a major barrier to entry for many scientists, who are more familiar with scripting-based languages like Python. Additionally, it means that Bonsai cannot readily integrate with most modern machine learning methods, which are often written in frameworks that utilize Python.
- 3) Mastering or learning ROS (for Heron) is quite difficult and therefore out of reach for many research scientists.
- 4) While *improv* can work across platforms, the Bonsai framework is currently limited to Windows Desktop operating systems only (<https://bonsai-rx.org/docs/articles/installation.html>), which restricts accessibility for users who prefer or are restricted to macOS or Linux systems. Although powerful in its own right, Bonsai does not offer full cross-platform support. While Heron has been tested under Linux and macOS, users struggle to run it on Windows (at the moment) (GitHub - S2-group/experiment-runner: Tool for the automatic orchestration of experiments targeting software systems).
- 5) While visual programming approaches can be powerful and intuitive initially (*cf.* <https://scratch.mit.edu/>), these software tools are inherently limited for flexible and sophisticated scientific investigations, as they do not allow for comprehensive control of the software and modeling. Therefore, Bonsai's visual programming approach may become too unwieldy for complex experimental designs, such as those we present in our *improv* study.

Together, while both Bonsai and Heron offer powerful capabilities for experiment orchestration, they each have specific limitations. We believe that *improv* serves as a flexible, scalable alternative, providing a more streamlined experimental experience for users who desire extensibility and interoperability, making an important contribution in this field.

While the authors state that such a comparison is not feasible, it is worth noting that, to the reviewer's knowledge, at least for Bonsai it is possible to conduct closed-loop experiments involving online modifications of states, rules, and parameters. Without a quantitative assessment of *Improv's* performance relative to existing technologies in the context of at least some of the proposed use cases, it is difficult to substantiate the claim of novelty.

We thank the reviewer for this comment and agree that such comparisons are highly useful for gauging the relative performance of these systems on various workloads. Before we show a direct comparison (below), we would like to note two caveats in comparing *improv* with Bonsai and Heron: First, as outlined above, the value proposition represented by *improv* does not exclusively (or even primarily) rest on performance. Second, we believe that *improv's* primary advantage is its flexibility as a system for orchestrating complex closed-loop experiments, particularly those involving user-specified models. This motivated us to demonstrate that *improv* is effective in a wide array of use cases, ranging from zebrafish calcium imaging and optogenetics to primate electrophysiology.

Figure R1: Caiman inside a Python node to run inside a Bonsai experiment.

To facilitate a direct comparison between Bonsai, Heron, and *improv*, we attempted to implement comparable versions of our streaming calcium imaging analysis pipeline shown in **Fig. 1** in both environments.

For Bonsai, we designed a calcium imaging experiment on a Windows computer (**Figure R1**, see **Fig. 1** in the main manuscript). Similar to the experimental design in **Fig. 1**, we imported prerecorded calcium data from a file, running each image through Caiman using similar code to our *improv Caiman Online actor* (**Fig. 1a**), and saved the processing time for each image. We leveraged Bonsai's Python Scripting library, a set of installable packages as an add-on to Bonsai, to run Caiman in a Python node inside Bonsai (**Fig. R1**). In principle, this experiment demonstrates a proof of concept that arbitrary Python code can be executed within the Bonsai framework.

For such a Bonsai implementation, we found that a Caiman Python node ran on averages at about 35 ms per frame, with rare large spikes into the seconds per frame processing time. Note that this reflects Caiman running inside Bonsai, not strictly the runtime speeds of Bonsai itself. *improv* and Caiman manage this in less than 30 ms (typically 10-20 ms, see **Supplementary Fig 2**), while **multiple other processes are running in parallel**; this is simply not possible in Bonsai in its current form.

In our experience, it is commendable that Bonsai nodes are highly separable and are designed as stateless transformations. For example, it is possible to convert an input double to an output float variable, or to grab an input image and render it in a popup window on the screen.

However, we encountered the following issues in implementing the closed-loop LNP model:

- 1) The main reason for this drawback in Bonsai is that only **one** Python process at a time may be run, and this process must be paused to extract data/information (ObserveOnGIL with a timer). Internal Python state is not maintained in other configurations.
- 2) Since Caiman is not coded in C#, we could not easily construct a native Bonsai node. Thus, we could not implement the same streaming or closed-loop paradigm, as all variables and data inside the Python node needed to be pre-loaded, and the Python node could not be exited until the entire dataset had been processed. That is, **while we could implement data processing in Bonsai, we could not readily implement a closed-loop design for online modeling** (compare **Figure 1f-g**).
- 3) Accumulating or modifying information necessary for experiments relying on multiple data points in the past is very difficult in Bonsai, as it relies on maintaining state or information across iterations of an input sequence. To run a Python node in Bonsai, an environment can be constructed via Module or Create Runtime nodes, and all subsequent Python scripts are executed in this environment. Yet to pass variables between nodes in a Bonsai graph requires (1) instantiation of all variables each time the node is run, (2) global variables, or (3) conversion to Bonsai and back to Python. The downsides of (1) or (3) is the possibly very costly initialization of importing libraries and

datatype conversions. The downside of (2) is that variables are now exposed outside their proper scope for use and are liable to name collisions. Nevertheless, we agree that this strategy could be effective in simpler, less computational heavy use cases.

- 4) Bonsai's Python function must be initialized and run anew **each time** it is called, meaning nothing with state or history can be maintained. Thus, Caiman, which requires multiple objects to persist, or any ML model with history, cannot be run in this manner. Nonetheless, it is possible that a different method that can facilitate iterative data passing appears to be possible with a PythonTransform node, but this is a script that does not maintain state. It is a single function that reads in a value (e.g., the image passed by a Camera node) and outputs a value (e.g., average intensity in the image).
- 5) Bonsai's brilliance of serial execution of programming instructions unfortunately also makes it challenging to develop software that can deal with the asynchronous, parallel nature of scientific data. A significant architectural limitation for distributed experiments is that the user cannot place multiple *nodes* on separate machines, leading in fact to the development of Heron (Heron: A Knowledge Graph editor for the intuitive implementation of Python-based experimental pipelines).

Nonetheless, recently, Bonsai has begun to consider supporting the dynamic updating of values while a Python script is running (<https://github.com/orgs/bonsai-rx/discussions/2025>; "*Get Dynamically Updating Values while Python Script is Running*", September 25, 2024). There is currently no supported functionality for this, although an example workaround was proposed that could allow global variables from within a Python script to be read by Bonsai, but not the reverse. That is, output from a script could be used for triggering, but input data between nodes or transform steps in a processing pipeline could not be used. We took this idea, modified it, and were able to demonstrate how a Python node running Caiman for image analysis could output scalar values to the command line (timing, average neural activity) (**Figure R2**). All data were read by the Python script during initialization of the runtime, Caiman ran on one image at a time internally during the evaluation node, and the result was dynamically read by Bonsai using a Timer every 0.5 seconds (simulated imaging rate).

Figure R2: (left) Async python example; (right, top) GIL warning; (right, bottom) Async timing with Caiman.

Unfortunately, this paradigm still uses a single Python process, meaning the **Python code execution must be paused** so that Bonsai can query it for outputs. We were unable to find examples or documentation of ways to dynamically supply the evaluation of a Python script with each new image at a time. **Thus, running more than 1 Python node in parallel is not currently possible in Bonsai, making all of our other *improv* experiments impossible to replicate.**

For insight into the current state of the package, a comment on the Bonsai forum (<https://github.com/orgs/bonsai-rx/discussions/2025>) suggests that the following asyncio approach is how model fitting might work: “...we use a similar approach for the *Bonsai.ML* package to fit model parameters asynchronously during optimization while still allowing the model to perform online inference without being blocked.” Yet we found no code examples as of this writing that demonstrate this capability; example code below shows Python code that does block and doesn’t receive data dynamically.

If we suppose that we do not use Python, and take better advantage of Bonsai’s parallelism, we are then restricted by the kinds of calculations currently supported (or must write our own in C#, a major barrier for many scientists). BonZeb is an example of software that used Bonsai for calcium imaging experiments

Figure Redacted

Figure R3: (From the BonZeb github webpage) Example of zebrafish imaging experiment.

(<https://github.com/ncguilbeault/BonZeb/tree/master/Examples/Calcium%20imaging>) and involved many steps that necessitated saving intermediary data to a csv file that was then read in by the next node. They triggered visual stimuli based on a local timer and analysis of tail beat frequency as their closed-loop paradigm, but we did not find any evidence of image analysis capabilities in a real-time closed-loop experiment.

In summary, we found that while Bonsai offers a rich and performant option for many use cases, it is simply not designed to accommodate complex Python code requiring parallelism or shared state, two of the key design features of *improv*.

Next, we attempted to implement the same routines in Heron. Unfortunately, we were unable to implement a meaningful direct comparison to Heron due to a lack of documentation of the preprint and the eLife-reviewed preprint, which means it is not yet published. The original Heron preprint states: *“We are not making public the specific experiment files since these are hardware specific and would need large changes to be made compatible with any other hardware.”* Moreover, we also could not find any information on runtime speeds in their manuscript. So, it might be possible that Heron can also produce comparable results, but it is difficult for us to compare *improv* to Heron directly. In comparison to Bonsai, which indeed has democratized simple workflows for systems neuroscientists, Heron has also not been implemented widely. Finally, while we agree that Heron might attempt similar goals, yet in the current version (*Reviewed Preprint v2February 24, 2025*, Heron: A Knowledge Graph editor for intuitive implementation of python based experimental pipelines, <https://doi.org/10.7554/eLife.91915.2>), they demonstrate Heron to be a possibly good tool to perform hardware integration and show screenshots to indicate that they might use “inference results to calculate angles of a cotton ball” (their Figure 4).

Nonetheless, we believe that our *improv* results not only show multiple, diverse examples of fully analyzed data sets using complex online analysis and modeling (**Figures 2-6**), but also offer comprehensively documented code for implementation, parameters for run times, modeling etc., and offer an attractive alternative to anyone trying to implement real-time experimentation.

Regrettably, in the absence of such evidence, I am unable to recommend this work for publication in Nature Communications at this time.

We hope that our comments and demonstrations above, coupled with our partial reimplementing of an *improvement* pipeline in Bonsai, have demonstrated that *improv* offers a performant and flexible alternative option for an important class of use cases in systems neuroscience worth sharing with the Nature Communications audience and beyond.

Reviewer #2 (Remarks on code availability):

NA

As we believe that our code quality and documentation are maintained by a professional computer scientist, and we are actively maintaining and updating the software, we hope this is a significant community contribution, as all actor examples and more are included in the code release.

Reviewer #3 (Remarks to the Author):

I stand by my previous comments, but the manuscript is in good shape. The authors highlight scenarios where analyses are only feasible during the experiment.

improv prioritizes ease of use and flexibility, and its design reflects this. Its reliance on an in-memory store is both its strength (it solves the coordination problem, which is indeed a big part of modern neuroscience experiments) and also its weakness (share-then-analyze is way more expensive than data reduction/analysis before message passing, especially now that in-memory stores no longer support zero-copy). It is good that the authors have committed to a maintenance strategy that does not rely on an obsolete in-memory store.

We thank the reviewer for this valuable feedback. We believe the updated software is now improved.

The software is open source, cross platform, and accessible, so now all that remains is to see how it will be adopted by the community.

We also hope that this tool will be helpful for neuroscience researchers who may not have considered such experiments.

09/10/2025

Point by point response to reviewers' comments:

improv: A software platform for real-time and adaptive neuroscience experiments

Black – Reviewer's comments and remarks

Blue - Our response to reviewers' comments and remarks

Red - Added or changed text to the revised manuscript

Again, we would like to thank the reviewers and editors for their extremely helpful suggestions and feedback throughout the review process.

Reviewer #2 (Remarks to the Author):

I thank the authors' for the comprehensive comparison of their software with established platforms such as Bonsai and Heron. Many of the points they raise are compelling; however, some seem to arise from limited familiarity with Bonsai rather than from intrinsic shortcomings of the platform itself. I therefore strongly encourage the authors to consider the option of including the key elements of this comparison in the manuscript, enabling readers to judge for themselves whether *improv* constitutes the best option for their experiments or whether existing alternatives are a better fit.

We thank the reviewer for this comment, and we fully agree with the reviewer's remark that Bonsai is an excellent software and likely will and should continue to serve the neuroscience community. However, we also think the *improv* will address complementary needs that require more complex workflows, and for users who prefer Python. We added additional text and changes to the discussion to explicitly point out the strengths of Bonsai, Heron, and other relevant software applications.

*For instance, BRAND, which likewise focuses on real-time experiments and shares many architectural features with *improv*, runs only on Linux, with core code written in C++ [63]. Similarly, Heron, which uses a graph structure to specify experimental flow and offers a GUI for interactive design of experiments, does not focus on performance and runs only on Windows [64]. Bonsai, which constructs experiments using a visual programming interface or custom C# scripts on Windows, is useful for simpler experiments but is less suitable for complex workflows or integration with modern machine learning methods that use Python [65]. Cleo, which targets closed-loop experiments, is focused on detailed biophysical simulation of the systems involved [38]. Moreover, while specialized systems like that of [33] have also established closed-loop setups in the zebrafish, these methods rely on specialized hardware and software solutions that are tightly matched to the specific system under construction.*

Reviewer #2 (Remarks on code availability):

Shared code is in good shape.

We thank the reviewer for this assessment.